# Combined Effects of Metakaolin and Hybrid Fibers on Self-Compacting Concrete

**DOI:** 10.3390/ma15165588

**Published:** 2022-08-15

**Authors:** Natalija Bede Odorčić, Gregor Kravanja

**Affiliations:** 1Faculty of Civil Engineering, University of Rijeka, Radmile Matejčić 3, 51 000 Rijeka, Croatia; 2Faculty of Civil Engineering, Transportation and Architecture, University of Maribor, Smetanova 17, 2000 Maribor, Slovenia; 3Faculty of Chemistry and Chemical Engineering, University of Maribor, Smetanova 17, 2000 Maribor, Slovenia

**Keywords:** self-compacting concrete, synthetic and steel fibers, metakaolin, rheology, mechanical properties, chloride penetration, SEM-EDS

## Abstract

There is a need to develop new construction materials with improved mechanical performance and durability that are low-priced and have environmental benefits at the same time. This paper focuses on the rheological, mechanical, morphological, and durability properties of synthetic and steel fiber reinforced self-compacting concrete (SCC) containing 5–15% metakaolin (M) by mass as a green replacement for Portland cement. Testing of the fresh mixes included a slump-flow test, density, and porosity tests. Mechanical properties were determined through compression and flexural strength. A rapid chloride penetrability test (RCPT) and the chloride migration coefficient were used to assess the durability of the samples. A scanning electron microscope (SEM) with energy dispersion spectrometry (EDS) was used to study the concrete microstructure and the interfacial transition zone (ITZ). The results show that a combination of metakaolin and hybrid fibers has a negative effect on the flowability of SCC. In contrast, the inclusion of M and hybrid fibers has a positive effect on the compressive and flexural strength of SCC. The fracture of SCC samples without fibers was brittle and sudden, unlike the fiber-reinforced SCC samples, which could still transfer a considerable load with increasing crack mouth opening deflection. Overall, the chloride migration coefficients were reduced by up to 71% compared to the control mix. The chloride reduction is consistent with the resulting compact concrete microstructure, which exhibits a strong bond between fibers and the concrete matrix.

## 1. Introduction

Concrete-based materials are constantly being developed with an aim to improve the mechanical performance and durability of structural concretes, as well as having environmental benefits at the same time. Self-compacting concrete (SCC) is widely used due to its main advantages over ordinary concrete, such as the possibility to fill the formwork even in the presence of dense reinforcement with no need for mechanical compaction, an easier concrete pumping procedure, shorter construction time, reduction in the amount of skilled labor needed onsite, no noise and vibration, and so on [1]. The main area for the application of SCC is members with dense reinforcement, jacketing of columns, precast panels, narrow and deep sections, underwater concreting, and decorative precast panels [2]. SCC typically contains the same basic components as ordinary concrete. The main difference is that amount of coarse aggregate is much lower, while the amount of cementitious materials is much higher in order to increase viscosity. Like ordinary concrete, SCC is characterized by brittle failure behavior in the hardened state. Thus, as already pointed out by many authors, the introduction of fibers into concrete is an effective means of improving its properties [3,4]. The fibers in concrete can greatly improve the toughness and resistance to cracking by acting as a bridge between the sides of the cracks and preserving the integration of the concrete up to high deformation [5]. The use of fibers in SCC improves crack resistance but may affect the workability of the concrete. According to a review of the literature, when a relatively small fiber volume fraction (up to 1%) is used, the workability of the resulting concrete should decrease within the acceptable range. On the other hand, even just a small addition of hybrid fibers can greatly improve the mechanical properties and durability of concrete. It is thus important to properly adjust the admixture dosage in combination with the proper mixing time to achieve the best results [6].

The incorporation of supplementary cementitious materials (SCM) in the SCC mixture is becoming an increasing necessity. In the past, the main motivation for using SCM was to reduce costs, but now—in addition to improving the properties of concrete—various SCM are also used as a partial replacement for Portland cement to reduce carbon emissions [7]. For example, high-ferrite Portland cement is used to prepare high corrosion resistance composites that are suitable for high-temperature marine environments [8]. Fly ash is widely used in hydraulic mass concrete structures and face slab concretes to reduce the shrinkage and risk of cracks in concrete [9]. Similar to fly ash, the utilization of electric furnace phosphorus slag has attracted growing attention in the construction of several hydraulic projects. The appropriate addition and fineness of phosphorus slag could improve the mechanical properties and durability of the concrete [10]. Another industrial by-product is ground granulated blast furnace slag (GGBS), which can be mixed with cement to reduce carbon emissions. Recently, GGBS has been shown to improve the workability, mechanical and durability characteristics of ultra-high performance concrete [11]. Among the many SCM that are used, metakaolin (M) is a material that has great potential with regard to the production of more sustainable concrete [12]. Metakaolin, Al_2_Si_2_O_7_, is an environmentally friendly pozzolanic material obtained by calcining kaolinite clay at relatively moderate temperatures of 650 to 750 °C [13]. Due to the high concentration of silica and aluminum, it is very reactive with calcium hydroxide, Ca(OH)_2_, and develops additional C-S-H gel, thus leading to an improved concrete microstructure and overall better performance of the concrete. Pozzolan metakaolin has been proven to enhance strength [14], permeability [15], chloride-ion diffusivity [16], and freeze and thawing resistance [17] in concrete materials.

A few studies have been published on SCC that include both pozzolans and fibers as reinforcement. For example, Mahapatra and Barahi [18] evaluated the multiple effects of hybrid fiber reinforced SCC with F-type fly ash and colloidal nano-silica. They found that a good correlation was obtained between the tested and predicted values of tensile strengths. Sivakumar et al. [19] examined the effects of metakaolin and alkali-resistant glass fibers on the performance of SCC. They found the glass fibers had no positive effects on the compressive strength of concrete or its resistance to chloride-ion penetration and water absorption. However, the presence of metakaolin and fibers in optimum proportions can improve the mechanical properties and durability of SCC. Ackay and Tasdemir [20] evaluated the effects of two pozzolans, namely silica fume and metakaolin, in steel fiber reinforced SCC. They concluded that the samples with metakaolin had a much higher fracture energy than those with silica fume, suggesting that the addition of metakaolin improves the bonding properties between the matrix and steel fibers. Abduljabbar et al. [21] assessed the mechanical effects of steel fibers on SCC made with microwave-incinerated rice husk ash, silica fume, and fly ash. They found that the tensile and compressive strength increased most when silica fume was used in combination with 2% steel fibers, based on the total weight of the binder. However, there has only been limited research on the properties of SCC with pozzolans and hybrid fibers, and the majority of the related studies are focused on only a few material properties at the same time. To the best of the authors’ knowledge, the combined effects of metakaolin with steel fibers and synthetic microfibers on the various properties of SCC have not yet been reported in the literature. Further, the findings reported in the literature on the behavior of SCC containing different amounts of M are not well documented with some inconsistencies in the results. Therefore, this work aims to investigate how the incorporation of M and both metakaolin hybrid steel and micro-synthetic fibers affect the rheological, mechanical, and chloride ion penetration properties of high-strength SCC. The morphology in the interfacial transition zone and the microstructure of the concrete were also investigated. In short, the work presented here makes an important contribution to our understanding of the comprehensive set of properties of SCC containing M and the combined effects of M and hybrid fibers.

## 2. Materials and Experimentation

To investigate the combined effects of metakaolin and hybrid fibers on the fresh and hardening properties of SCC, eight concrete mixes were developed with a different mass percentage of metakaolin and the same amount of hybrid fibers. The experimental part of the study was carried out at the Faculty of Civil Engineering in Rijeka, Croatia.

### 2.1. Materials

The typical ingredients used in the production of SCC are cement, a mineral admixture, coarse aggregate, fine aggregate, water, viscosity modifier, and superplasticizer. In the current study metakaolin (M) was also added to improve the workability, strength, and durability of SCC as SCM. According to previous studies of SCC containing M, the maximum amount which guarantees an increase in compressive strength is 15% of M by weight of cement [22]. Synthetic microfibers and steel fibers were also added to the concrete mixture to improve the mechanical properties of SCC. The improved properties of mixtures with steel fibers are increased flexural strength and ductility. Synthetic microfibers offer many advantages when used in concrete, such as reducing segregation and plastic shrinkage cracking caused by moisture loss at the concrete’s surface, and increasing the uniformity of mixing water, which contributes to better surface finishing.

Portland cement (PC) CEM II/A-LL 42,5 R with high initial strength conforming to standard HRN EN 197-1 was used as the binder in all mixtures. The content of PC was 80–94% Portland clinker and 6–20% additives. Locally available mineral additive M was used as a partial replacement for cement in SCC at 5%, 10%, and 15% by weight of cement. Table 1 summarizes the physical and chemical properties of the PC and mineral additive used in the studied mixtures.

Crushed limestone aggregates with a density of 2740 kg/m^3^ and a maximum grain size of 8 mm were used in the mixture preparation. The ratio of coarse aggregates to sand was kept constant and equal to 0.54:1. Dynamo SF 16S, a polycarboxylate superplasticizer for fresh concrete with low viscosity, was used in the production of SCC. Dynamo SF16 S has the best dispersing effect if added after the cement, admixtures, mineral additives and at least 80% of water are mixed. The recommended dosage is 0.2–2% in relation to the cement weight. The density of the superplasticizer used was 1 g/cm^3^ and chloride content was <0.1%. Viscosity modifying admixture (VMA) was added to control the rheology of the mix and increase cohesion and segregation resistance. Hooked-end steel fibers type DE 30/0.55 N, made by KrampeHarex, were used for hybrid SCC. The nominal diameter of the fiber is d = 0.55 mm and the nominal fiber length is L = 30 mm (aspect ratio L/d = 55). The tensile strength of the fibers reported by the manufacturer is 1350 N/mm^2^. The typical practical minimum dosage is 20 kg/m^3^. Synthetic fibers used in the SCC mixtures were the high-performance fibrillated microfibers Fibrofor High Grade 190 produced by Contec Fiber. The fibers are made of pure polyolefin with a bulk density of 0.91 g/cm^3^ and are environmentally friendly. The diameter of the microfibers is less than 0.03 mm, and the length of the fibers is 19 mm. The tensile strength was reported by the manufacturer as 400 N/mm^2^. For fibrillated microfibers, the typical dosage guideline is 1 kg/m^3^.

### 2.2. Mix Design

For this experimental program, eight concrete mixes were made. The mixes were prepared with different replacements of the cement portion with metakaolin (up to 15% of the cement mass) and the same amount of hybrid fibers.

The detailed proportions of the eight SCC mixes are listed in Table 2. All SCC mixes have the same proportions of fine aggregate, coarse aggregate, water, superplasticizer, and stabilizer. The dosage of superplasticizer and VMA were constant in all mixes, at 2% and 0.6% by cementitious material weight, respectively. Moreover, the total amount of cementitious material (500 kg/m^3^) and water-to-binder ratio (w/b = 0.4) were also kept constant for all mixes.

In the first four mixes (SCC-0M, SCC-5M, SCC-10M, and SCC-15M) only the amounts of cement and metakaolin were varied. Hence, the first mixture was the SCC control mixture denoted as SCC-0M. In the next three mixtures (SCC-5 M, SCC-10 M, and SCC-15 M), a certain amount of cement was replaced with M (by 5%, 10%, and 15% of the mass of cement). The influence of the effects of metakaolin and hybrid fibers was studied by adding 50 kg/m^3^ of hooked end steel fibers and 1 kg/m^3^ of synthetic microfibers in the next four mixes (HySCC-0M, HySCC-5M, HySCC-10M, and HySCC-15M). The main aim of this experimental investigation was to study the combined effects of M and hybrid fibers on the various properties of SCC and not to find the optimum content of hybrid fibers which results in the maximum improvement of mechanical properties. For this reason, the amount of synthetic and steel fibers was selected according to the recommendations given in engineering practice. Additionally, to better capture the influence of the combined effects of metakaolin and hybrid fibers on various properties, all other mix components, except the amount of M, were kept constant. Therefore, only one volume fraction of synthetic and steel fibers was considered. When hybrid fibers are neglected, the four mixes without fibers correspond to mixes SCC-0M, SCC-5M, SCC-10M, and SCC-15M without fibers.

When placing the concrete in molds only hand compaction was provided. For each concrete mix, the same test specimens were prepared: cubes with 150 mm sides to determine compressive strength, beams 100 mm × 100 mm × 400 mm to determine flexural tensile strength and fracture behavior, and cylinder samples with 100 mm diameter to determine chloride ion penetration resistance.

### 2.3. Rheological Properties of SCC

The properties of fresh concrete mixes were tested according to current standards for SCC used in Europe. Testing of fresh SCC was carried out immediately after mixing had been completed and within a period of a maximum of 30 min to avoid losing the workability of the mixture. The various fresh properties of each SCC mixture with the same composition were measured two times.

The properties required for the self-compatibility of concrete were characterized by testing the fresh mixtures, including with a slump-flow test method according to HRN EN 12350-8 [23] and a V-funnel test method according to HRN EN 12350-9 [24].

The flowability of SCC was determined by the slump flow test method by measuring the spread diameter and slump-flow time (T_500_ value). The slump value diameter for SCC concrete ranges between 550–850 mm, according to HRN EN 206 [25]. Satisfactory flowability is one of the most important parameters since it affects the filling ability of fresh concrete and then the properties of the hardened concrete, as well as the variability of test results. Viscosity in SCC is related to the speed of flow in the slump-flow test until reaching the diameter of 500 mm (T_500_) or the efflux time in the V-funnel test. According to the European guidelines for SCC [26], typical workability requirements for slump flow time T_500_ are in the range of 2–5 s, and for V-funnel flow time are between 6–12 s. The time value does not measure the viscosity of SCC but it is related to it by describing the rate of flow. Concrete with low viscosity will have a very quick initial flow and then stop. In contrast, concrete with high viscosity may continue to creep forward over an extended time. Additionally, segregation and bleeding of fresh concrete were visually observed during the slump flow test and V-funnel test at T_5min_. Further, the density and porosity of the fresh concrete mixes were also measured. The air content of the concrete mixes was estimated according to the standard HRN EN 12350-7 [27] using the pressure gauge method. The test procedure described in HRN EN 12350-6 [28] was used for determining the density of the freshly mixed concrete.

### 2.4. Mechanical Tests

The design of elements and structures made of hybrid fiber reinforced SCC requires knowledge of its basic mechanical parameters. Compressive and flexural tensile tests were performed to evaluate the mechanical properties of SCC blends containing metakaolin. For each material property, at least three specimens were tested to verify the reliability of the test results. Until the beginning of each test, the samples were stored in a water tank at 20 °C. The tests were performed on specimens that were 28 and 56 days old. For the compression test, a universal testing machine with a maximum load capacity of 3000 kN was used. This test was also performed on specimens that were 28 and 56 days old. To determine the concrete´s compressive strength, cube specimens with sides of 150 mm were used, according to the testing standard HRN EN 12390-3 [29]. To get insight into the flexure behavior of the concrete mixes, prismatic beams with a cross-section of 100 mm × 100 mm and a span length of 300 mm without a notch were tested in a three-point bending test with displacement control. The flexure tests were carried out using a Zwick Roell universal testing machine with a load cell of 50 kN. To investigate the post-peak response and fracture behavior, the load-displacement history was recorded during testing—for SCC without fibers until fracture and for HySCC with fibers up to deflection of at least 4 mm.

### 2.5. Durability Tests

#### 2.5.1. Rapid Chloride Penetrability Test and Electrical Resistivity

The rapid chloride penetrability test (RCPT) in terms of total charge passed through concrete samples after 28 days of water curing was measured according to ASTM C 1202 [30]. If the total charge passed is >4000 C chloride ion penetrability is high, moderate if ranging from 2000–4000 C, and low for range 1000–2000 C. The permeability test was performed on cylindrical specimens with a diameter of 100 mm and a height of 50 mm. One end of the sample was in contact with a sodium chloride solution (NaCl) and the other with a sodium hydroxide solution (NaOH). A DC potential of 60 V was applied to the samples’ surfaces, and the total continuous charge was measured with the Model 58-E5204 control device for a total duration of up to 6 h.

To eliminate the effects of heat on ion mobility and final Coulomb values in concrete samples with fibers, a modified ASTM C 1202 test was used [31]. Linear extrapolation up to 6 h was performed by multiplying the 30 min values by 12 to account for heat effects associated with the increased electrical conductivity of the fibers. For up to 30 min the temperature gradient did not increase by more than 10 °C. To enhance the reliability of the results, we performed additional measurements of the chloride penetration depth (see Section 2.5.2).

The electrical resistivity for all mixtures corresponding to the electrical current measurements was calculated using Equation (1):(1)$$\mathrm{ER}=\frac{UA}{IL}$$
where ER is electrical resistivity (Ωm), *U* is the absolute value of the applied voltage (*V*), *A* is the cross-sectional area of the sample (m^2^), *I* is the current at 30 min after the voltage is applied (*A*) and *L* is the thickness of the sample (m). The length of the corrosion propagation stage until spalling of the concrete can be established by a linear correlation between the inverse concrete ER and the steel corrosion rate in µm Fe per year.

#### 2.5.2. Chloride Migration Coefficient from Non-Steady-State Migration Experiment

In addition, chloride penetration was evaluated by calculating the chloride migration coefficient from the non-steady-state migration experiment. Similar to the RCPT, the cylindrical samples were kept under a potential difference between a cathode solution and anode solution according to the standard. After completion of the test, all tested samples were axially split into two pieces of fractured surfaces and sprayed with 0.1 M AgNO_3_ solution. Chloride penetration depth was assessed by a colorimetric method. The average chloride penetration depth was determined on the concrete surface by a change in color in the presence of chloride ions, which chemically led to the formation of white AgCl product. The chloride migration coefficient was calculated using Equation (2) [32]:(2)$$D_{nssm}=\frac{0.0239\left(273+T\right)L}{\left(U-2\right)\cdot t}\left(x_{d}-0.0238\sqrt{\frac{\left(273+T\right)Lx_{d}}{U-2}}\right)$$
where *D_nssm_* is the non-steady-state migration coefficient (×10^−12^ m^2^/s), *U* is the absolute value of the applied voltage (*V*), *T* is an average value of the initial and final temperatures in the anolyte solution (°C), *L* is the thickness of the specimen (mm), $x_{d}$ is an average value of the penetration depth (mm), and *t* is the test duration in hours.

### 2.6. SEM and EDX Sample Preparation and Morphology Examination

Cured SCC samples at the age of at least 28 days were smashed into pieces less than 1 cm^3^ in size and placed in 96% ethanol to replace the pore solution and then dried in the ventilation oven at room temperatures. A precision etching and coating system (Gatan PECS II Model 685) with dual ion sources for the etching of solid samples was used for concrete coating with Pt. A JSM-7800F field emission scanning electron microscope equipped with (EDX) energy dispersive spectrometry was used to examine at different magnifications the microstructure of SCC samples and the interfacial transition zone (ITZ) between aggregates, binder pastes, and reinforced fibers.

## 3. Results and Discussion

This section presents the results of the rheological, mechanical, and chloride permeability tests performed, as are those of the microstructure analysis on the bonds between the fractured SCC specimens and fibers.

### 3.1. Rheological Properties of SCC

Visually, all mixes were found to be homogeneous with no evidence of segregation or significant bleeding of the cement paste. As seen in Figure 1, the SCC mixes become less workable as the M percentage increases and show a reduction in slump flow diameter (D_max_) and longer flow time (T_500_) as compared with SCC with 0% M. This shows that concrete with M has a higher water demand. These results are in good agreement with studies by Vejmelková et al. [33]. For SCC mixes without fibers, the flow diameters were in the range of 750–650 mm. The test results show that all SCC mixes without fibers meet the flowability requirements according to the European guidelines [26]. The time for fresh mixtures to reach 500 mm flow was in the range of 3–9 s for all SCC mixtures without fibers. Moreover, a combination of metakaolin and hybrid fibers had an additional negative impact on the workability of SCC by reducing slump flow diameter even more, to less than the minimum required slump flow diameter of 550 mm, and by significantly increasing the flow time T_500_ in the flow test with increasing M content, up to 27 s.

From these results, it can be concluded that the reduction in flowability with increasing the proportion of metakaolin is greater in mixes with hybrid fibers than in mixes without them. The test results show good consistency and acceptable scatter in the case of a small amount of M. However, the scatter of test results for a specific concrete mixture is relatively high, especially for cases with a high content of M. In addition, it can be observed that the addition of fibers not only reduces slump flow diameter but also increases the scatter of test results. This can be attributed to the inhomogeneous distribution of fibers in the concrete matrix causing greater heterogeneity inside the mixture.

The effect of metakaolin and hybrid fibers on the viscosity of fresh SCC is even greater. The rate of flow for SCC with metakaolin and hybrid fibers is much slower compared to the reference mixture. The V-funnel flow time was also measured. The time required to flow through the V-funnel apparatus increases along with the M content, ranging from 8–14 s for SCC mixes without fibers, while for all HySCC mixes flow was blocked due to the fibers. Consequently, concrete mixes containing hybrid fibers and 10% to 15% of M replacing cement by weight do not show the characteristics of the fresh SCC mixtures.

The porosity of SCC mixes increases slightly with increasing the M content (Figure 1b). However, the porosity increased significantly for the HySCC mixture with 15% of M replacing concrete by weight. The increased viscosity and thus reduced flowability of the HySCC-15M mixture are the main reasons for the high porosity. The increased porosity is also a consequence of inhomogeneous fiber distribution in fresh concrete during the casting process, causing fibers to become trapped and generating more pores.

From Figure 1b,d, it can be seen that the porosity is directly related to the fresh density. When the porosity decreases slightly, the density increases accordingly. Further, Figure 1d shows some interesting results. It is observed that replacing PC with M in the concrete mixtures had very little effect on the density. However, SCC mixes with 5% and 15% of M replacing cement by weight give a slightly higher density than SCC without M and SCC with 10% M. It could be concluded that the viscosity affects the density of SCC mixes with M, and that mixtures with a higher viscosity have slightly higher density since the creation of pores is reduced. As expected, for mixtures with approximately the same viscosity those with a higher porosity will have a lower density. Further, for mixtures with the same porosity and viscosity, mixtures containing hybrid fibers have a higher density.

Hence the rheological properties of SCC mixed with a combination of M and hybrid fibers are characterized by relatively high viscosity and cohesion. On the other hand, incorporation of M and fibers tends to reduce flowability, and this is most likely caused by the high reactivity of the M.

### 3.2. Hardened Concrete Properties

#### 3.2.1. Compressive and Flexural Strength

Figure 2 shows the effects of metakaolin as well as those of a combination of metakaolin and hybrid fibers on the compressive and flexural strength of SCC at 28 days. As can be observed from Figure 2a, there is a general increase in compressive strength as the M content increases for mixes both with and without fibers. In the case of the SCC mix with 10% M, a slight decrease in strength is observed. Since the strength of concrete is directly related to the density of the concrete matrix, it can be concluded that a lower density is the main reason for this decrease (Figure 1d and Figure 2). The compressive strength values increase from 16–29% for the SCC mixes without fibers and from 16–21% for the SCC mixes with fibers. The results show that even a small admixture of M makes a significant contribution to the increase in compressive strength.

In general, inclusion of hybrid fibers has a positive influence on the compressive strength of SCC. The SCC mix with hybrid fibers, HySCC-0M, shows about 9% higher compressive strength than SCC without fibers, SCC-0M. However, as the M content increases, the incorporation of the fibers becomes less significant. It can thus be concluded that the incorporation of M contributes more to the compressive strength than hybrid fibers. The SCC mixes with M and hybrid fibers showed an increase in compressive strength in the range of 9–31% compared to the SCC mix without fibers, SCC-0M. Consequently, the concrete mix with hybrid fibers and the highest percentage of M achieved the highest compressive strength, up to 31% higher than SCC-0M without M and fibers. In addition, Figure 2b shows that the compressive strength of SCC mixed with metakaolin at the age of 28 and 56 days show the same trend, higher compressive strength with a higher amount of M as a consequence of the pozzolanic reaction. The compressive strength of SCC with 5–15% of M increased faster at the age of 28 days than at 56 days. The compressive strength of SCC with M at 28 days increased up to 29% compared to SCC-0M, while after 56 days it only went up another 14%. This is related to the higher reactivity of the blended binder with M [19].

From Figure 2c, it can be seen that the addition of hybrid fibers has no significant effect on the flexural strength, and on average the addition of hybrid fibers only slightly improves the flexural strength values. However, fiber-reinforced SCC blends with 15% M show about 15% higher flexural strength values compared to SCC without M. This indicates that a small addition of M leads to a better bonding of the fibers with the matrix [20]. The hardened density results shown in Figure 2d indicate that all the mixes in the study meet the density requirements for normal concrete as specified in HRN EN 206 [25], indicating a slight reduction in density with an increase in M for mixes both with and without fibers. As expected, the density of the HySCC mixes with fibers is slightly increased compared to the same SCC mixes without fibers.

#### 3.2.2. Influence of Fibers on Displacement Fracture

As can be seen in Figure 3, Figure 4 and Figure 5, the damage to the SCC prism without fibers happened suddenly and the fracture was brittle, unlike the fiber-reinforced SCC samples, which were still able to transfer a considerable load with increasing crack mouth opening deflection. The letters in Figure 3, Figure 4 and Figure 5 denote the number of tested samples for each SCC mix. These are ordered in such a way that the letter A denotes the first tested sample and H the last one. The hybrid fibers in SCC increased its overall performance in the terms of impact endurance, ductility, and toughness. The practicality of using hybrid fibers in concrete is clarified by Singh et al. [34]. They concluded that the highest values for almost all residual strengths for the hybrid blend were obtained by combining two different fiber types. The addition of steel fibers affects the hardening and softening behavior, while synthetic fibers tend to reduce plastic shrinkage cracking [35]. Since the dosage of fibers in our study is the same, the key factor that affects the performance of the fiber-reinforced SCC mixtures is the matrix properties and bond strength between the fibers and matrix. The addition of pozzolans such as M resulted in better pore refinement and higher fracture energy [36].

In the case of several SCC mixtures, it was clear that some of the bent fibers did not disperse which resulted in the uneven distribution of fibers. This may be attributed to the fact that the viscosity of the SCC mixtures was very significant, which negatively affected the fiber distribution and orientation inside fresh concrete [37]. The reason for large variations in the results of individual samples may be related to the different numbers of steel fibers in the cross-sections of the crack [38]. For example, HySCC-0M prisms from the same batches have both hardening and softening straining. In the case of hardening behavior, the fibers were more vertically oriented relative to the bending loading direction, leading to enhanced mechanical properties. While it is evident that samples with softening behavior have fewer fibers in the cross-section that is more horizontally oriented (Figure 4).

### 3.3. Durability Test

The rapid chloride permeability test (RCPT), concrete electrical resistivity, and chloride bulk diffusion test from non-steady-state migration were conducted to evaluate the performance of SCC containing metakaolin and fibers against chloride attack and chloride-induced corrosion.

#### 3.3.1. Rapid Chloride Penetrability Test

The RCPT was performed in accordance with ASTM C1202, and the results were calculated as the average charge passed through at least four specimens (Figure 6a). The linear effect of chloride reduction is observed when the percentage of M is increased from 5% to 15% of the cement weight. In the case of SCC samples without fibers, the percentage reduction in RCPT compared to the control mixture was 37%, 58%, and 84% when 5%, 10%, and 15% of M were used, respectively. Unlike the control SCC-0M, all M blended SCC mixtures without fibers fall under the category of a medium-to-low probability of chloride penetration. The resistance of chloride ions in SCC samples depends on the concentration of OH^−^ ions located in the pore solution and on the composition of mineral admixtures. Since the alkali content in Portland cement is responsible for pore fluid transportation in concrete, changing the composition with a small dosage of M influences the curing regime, as the OH^−^ in the pore fluid and particle packing density together enhance the resistance against chloride [39].

As expected, chloride penetration increases when hybrid fibers are used. The congestion of fibers in a concrete mixture affects the self-compacting ability, which increases the pore structure and consequently the RCPT values [40]. Another reason for the increased chloride penetration is the high electrical conductivity of the HySCC samples. Nevertheless, the pozzolanic reaction of M decreased the RCPT values for HySCC by forming more resistive paths in the concrete microstructure and improved the bonds between fibers and hardened cement paste. In the case of HySCC samples with fibers, the percentage reduction of RCPT compared to the control mix was 16%, 30%, and 50% when 5%, 10%, and 15% M were used, respectively.

The mixtures show a decrease in total charge passed for samples both with and without M content from 28 up to 56 days (Figure 6b). This can be explained by the hydration and formation of a denser microstructure over time [41]. In addition, the total cement content and the use of SCMs play an important role in the hydration and penetration of concrete, with SCMs delaying hydration so that more time is needed to develop the final permeation resistance [42].

The SCC electrical resistivity values support the findings from the RCPT. The resistivity values increased with the increasing M content, from 16 Ωm up to 92 Ωm without fibers, and from 7 Ωm up to 27 Ωm with fibers (Figure 6c). Replacing a small amount of Portland cement with M resulted in enhanced protection of SCC samples against corrosion. As can be seen in Figure 6d, the corrosion rate of µm Fe per year decreases significantly with increasing the M content, indicating that M is beneficial in protecting steel bars or fibers [31]. Concrete with a high passing charge (Coulomb) does not necessarily have very high chloride diffusivity. This becomes even more obvious when using fibers that conduct electricity. Therefore, it is useful to perform a chloride bulk diffusion test, which is less dependent on the electrical conduction of the SCC samples.

#### 3.3.2. Chloride Migration Coefficient from Non-Steady-State Migration

Figure 7 shows the results for the chloride migration coefficient from non-steady-state migration for SCC samples without fibers and HySCC samples containing fibers. Replacing cement with M results in a significant decrease in the chloride migration coefficient, D_nssm_ (×10^−12^ m^2^/s). For SCC samples without fibers, the chloride migration coefficient decreased from 49.98 × 10^−12^ m^2^/s to 14.53 × 10^−12^ m^2^/s when the M content was increased from 5% to 15%. Compared to the control mixture, the chloride migration coefficient was reduced by up to 71%. The calculated chloride migration coefficient appears to decrease exponentially with higher M content. Similar to the findings of Badogiannis et al. [43], it can be concluded that increased M content in SCC samples has a positive effect on increasing compressive strength and reducing the chloride migration coefficient. As mentioned above, fibers have a negative effect on chloride permeability, and the values of the chloride migration coefficient are higher than in specimens without fibers.

Figure 7b,c shows the average depth of chloride migration in mm on tested specimens that were axially split into two pieces and sprayed with a 0.1 M AgNO_3_ solution. With the improved and more compact concrete structure, due to the pozzolanic reaction of M, the average depth of chloride migration decreased from 41.50 mm down to 12.70 mm.

### 3.4. Morphology

#### 3.4.1. SEM Analysis

To acquire a sense of the concrete microstructure and the bonding degree between the hardened 28-day-old SCC and the synthetic and steel fibers, a scanning electron microscope (SEM) analysis was carried out (Figure 8). Figure 8a,b shows micrographs of a fractured SCC sample fixed with synthetic and steel fibers. Examination of the interface between the synthetic fiber and the concrete matrix shows that there is a very good bond between the fiber and solid concrete surroundings. An acceptable level of bonding is also observed between the steel fibers and the concrete matrix, with the concrete adhering strongly to the surface of the steel fibers even after the fracture. In addition, neither fiber chemically affected the microstructure of the fabricated SCC composites. The findings are compatible with a study by El-Dieb et al. [44], who examined the microstructural characteristics of ultra-high-strength SCC incorporated with steel fibers.

Quantitatively, the most important hydration product and most important factor affecting the strength of the hardening mass of a concrete sample is the hydrated calcium silicate phase (C-S-H), which has a recognizable microstructure in the form of a “honeycomb” structure [45]. In the test sample SCC-10M with 10% of M, it is evident that C-S-H gel is combined with crystallization products like ettringite and calcium hydroxide Ca(OH)_2_ (Figure 8d). Ettringite has a crystal microstructure in the form of thin needles, while Ca(OH)_2_ has a morphology in the form of hexagonal plates. Both crystallization products help form a denser concrete microstructure with more filled pores.

#### 3.4.2. SEM-EDS Mapping

SEM-EDS and mapping test was performed to visually assess the different chemical composition on the concrete´s surface and the bond at the interface between the fibers and concrete matrix. Figure 9a shows the results of SEM-EDS mapping at a surface of a fractionated hySCC-10M sample with the identification of elements O, Ca, C, Si, K, and Al. As seen in Figure 9b the surface of synthetic fiber is smooth and well adhered to the concrete matrix. As expected, the main chemical composition of the synthetic fiber is C and O, constituting the whole polymer structure. Typical chemical elements of concrete such as Ca, O, C, and traces Si and Al are found in the interfacial transition zone. In the case of the SCC sample with steel fibers, the chemical composition showed that the main element of the steel fibers was Cu, followed by Zn and Fe. Around the alloy fibers, there is an adherent concrete material with the typical chemical composition containing elements such as Ca, C, O, and Si (Figure 9c).

## 4. Conclusions

The results of a comprehensive examination of a set of properties of fresh and hardened SCC mixes containing metakaolin and hybrid fibers fixed at only one volume fraction show that it is possible to obtain SCC with increased strength and durability. The conclusions of this study can be summarized as follows:SCC mixtures containing metakaolin and hybrid fibers require the addition of a larger amount of superplasticizer than SCC without fibers and M to ensure adequate self-compacting properties. With an increase in M, the flowability of SCC mixtures decreases slightly, while the viscosity and cohesive strength increase.Compressive strength generally increases with the increase in M content for both mixtures with and without fibers. The highest compressive strength is obtained for the SCC mix with hybrid fibers and the highest percentage of M, up to 31% higher than SCC without M and fibers. In addition, fiber-reinforced SCC mixes with 15% M have approximately 15% higher flexural strength values than SCC without M.To obtain better mechanical properties, the most important parameter for SCC is the distribution and orientation of the fibers. In the case of several SCC mixtures, it became clear that some of the bent fibers did not disperse, resulting in an uneven distribution of the fibers.The linear effect of chloride reduction is observed when the percentage of M is increased from 5% to 15% of the cement weight. Consequently, the calculated chloride migration coefficient appears to decrease with higher M content.The electrical resistivity increases with the increase in M content from 16 Ωm to 92 Ωm without fibers and from 7 Ωm to 27 Ωm with fibers.The microstructural investigation of the concrete reinforced with hybrid fibers confirmed the assumption of a strong bond between the fibers and concrete matrix. The crystallization products form a denser concrete structure with more filled pores.The test results emphasize the importance of the rheological properties of SCC such as adequate workability to ensure improved mechanical performance and durability. Hence, it is recommended to study the combined effects of metakaolin and hybrid steel and synthetic fibers at various volume fractions.

Such high-performance SCC shows great potential for use in construction applications where higher crack resistance and resistance to chloride penetration are required.

## Figures and Tables

**Figure 1 materials-15-05588-f001:**
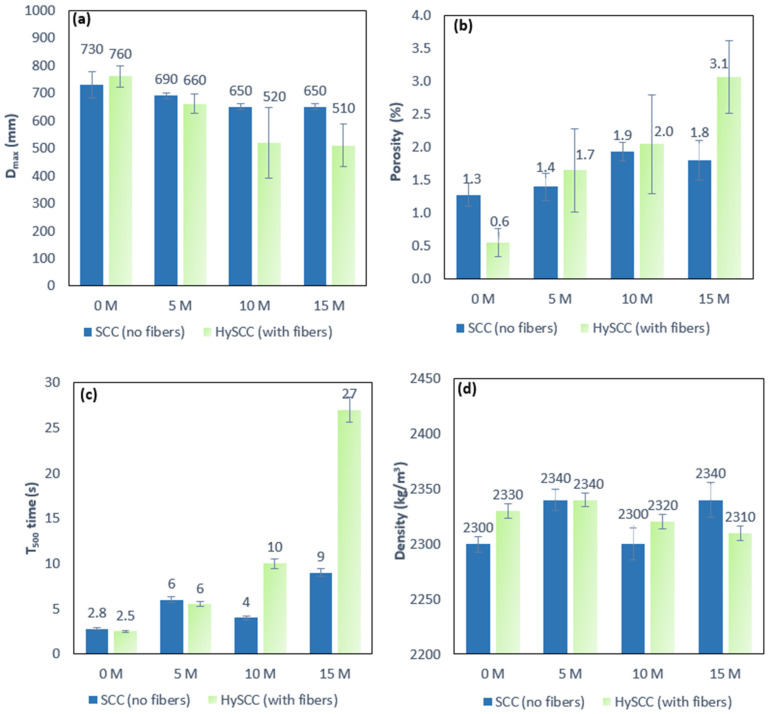
Effects of metakaolin and hybrid fibers on the workability of fresh SCC mixes on (**a**) flow diameter, (**b**) porosity, (**c**) time of flowing, and (**d**) density.

**Figure 2 materials-15-05588-f002:**
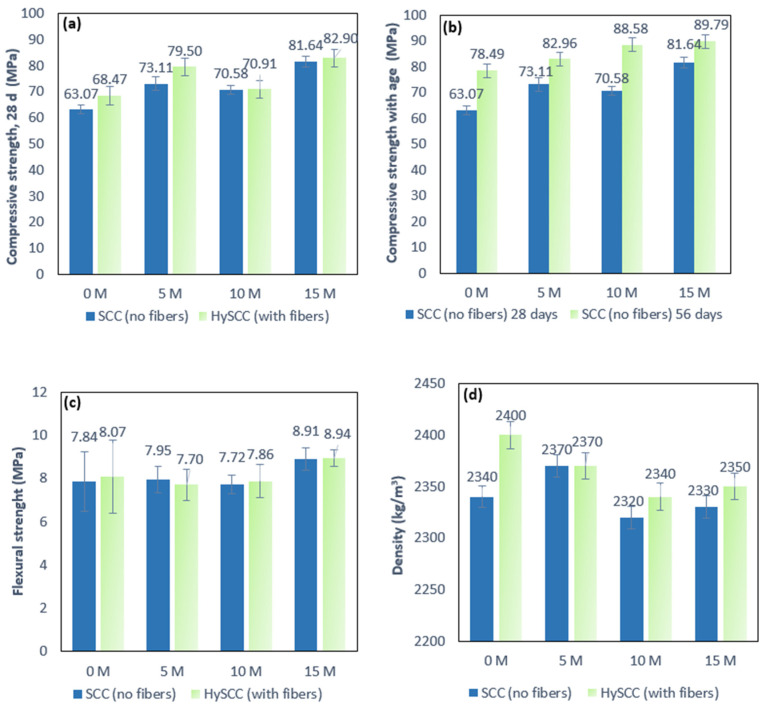
Effects of metakaolin and hybrid fibers on (**a**,**b**) the compressive strength of SCC at 28 days and 56 days, (**c**) flexural strength at 28 days, and (**d**) density in a hardened state.

**Figure 3 materials-15-05588-f003:**
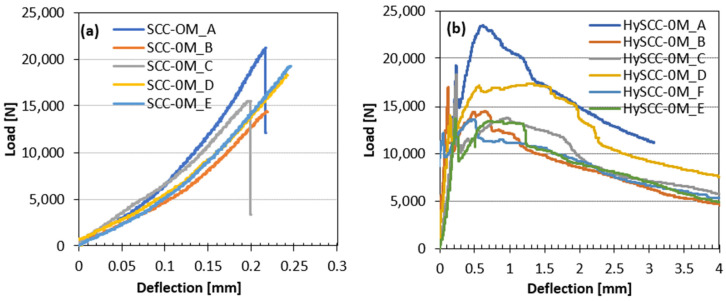
Load–deflection curves of (**a**) SCC-0M without hybrid fibers and (**b**) HySCC-0M with hybrid fibers.

**Figure 4 materials-15-05588-f004:**
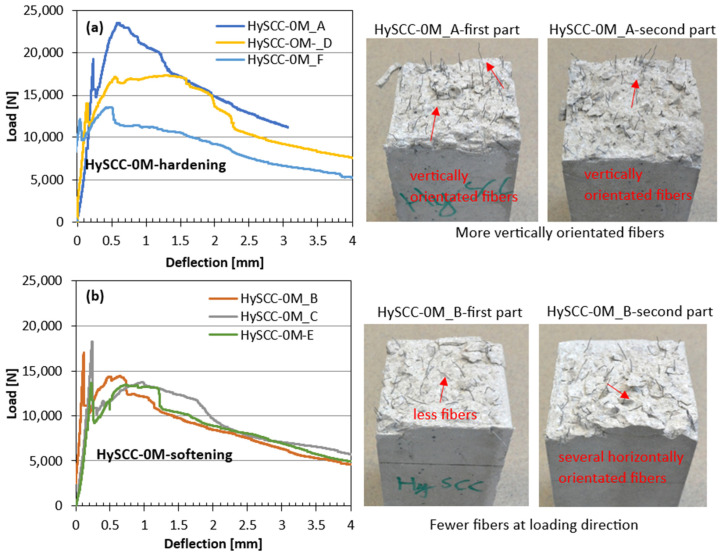
Load–deflection curves for HySCC-0M, where orientation of added steel fibers influences on the (**a**) hardening or/and (**b**) softening straining behaviors.

**Figure 5 materials-15-05588-f005:**
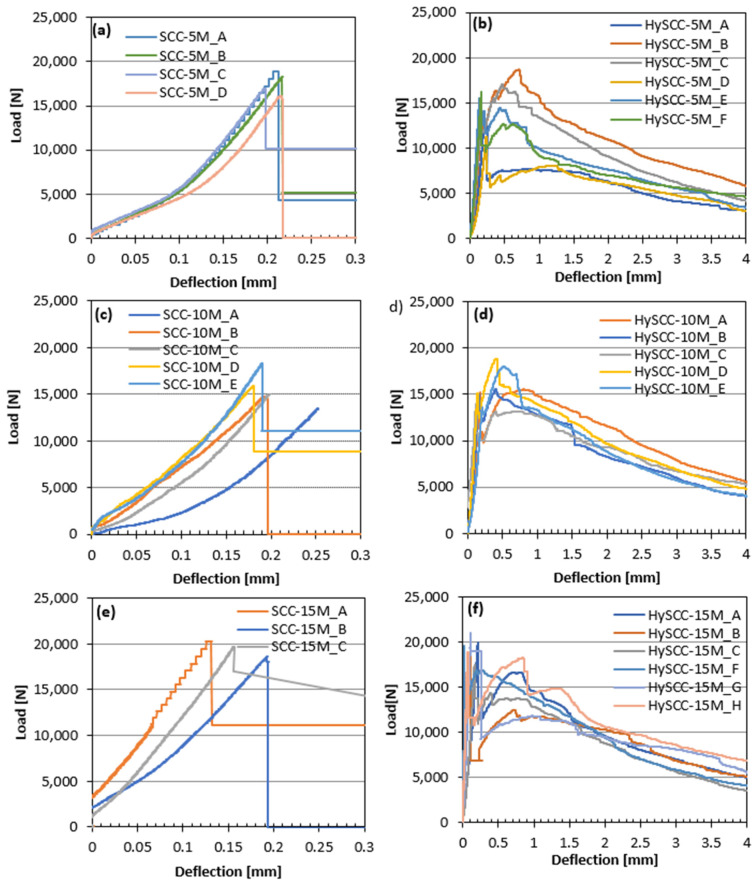
Load–deflection curves for: (**a**) SCC-5M without fibers, (**b**), HySCC-5M with fibers (**c**), SCC-10M without fibers (**d**), HySCC-10M with fibers (**e**), and SCC-15M without fibers (**f**) HySCC-15M with fibers.

**Figure 6 materials-15-05588-f006:**
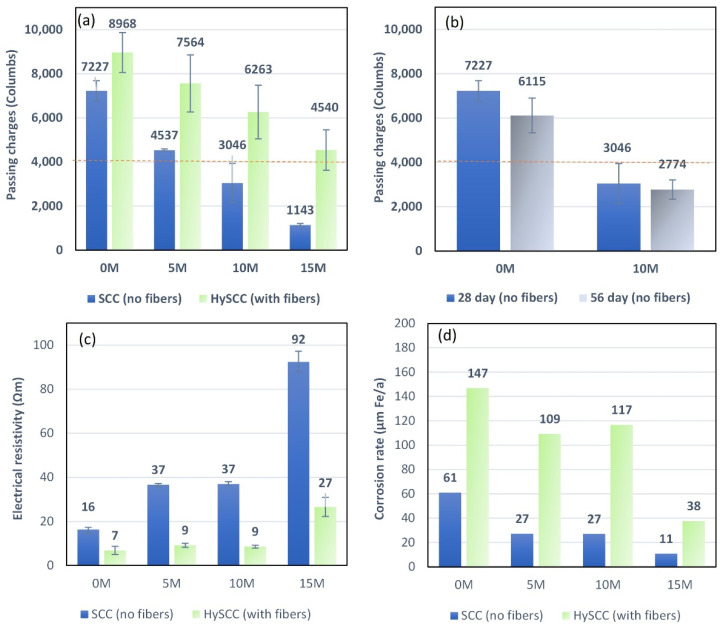
Influence of M content on (**a**) total charge passing in RCPT for the SCC and HySCC samples, (**b**) total charge passing in RCPT for the SCC samples at 28 and 56 days, (**c**) electrical resistivity of the SCC and HySCC samples, and (**d**) corrosion rate in µm Fe per year for the SCC and HySCC samples.

**Figure 7 materials-15-05588-f007:**
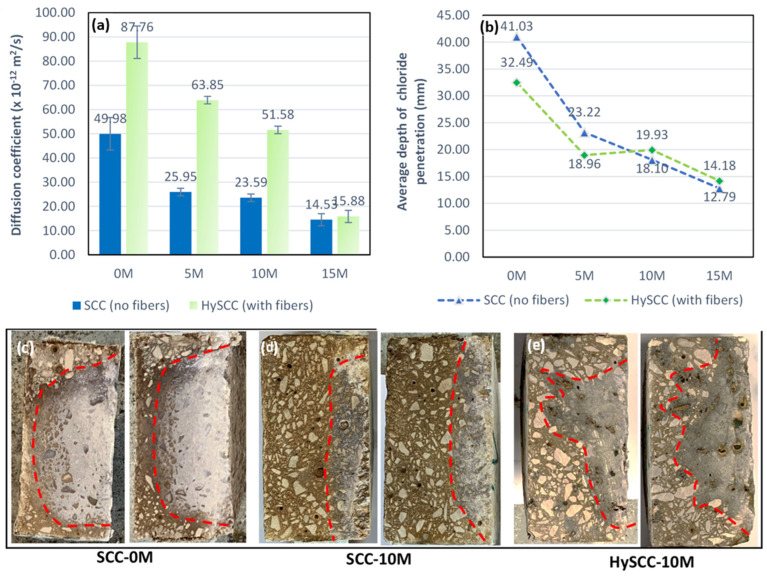
(**a**) Chloride migration coefficient from non-steady-state migration for SCC samples without fibers and HySCC samples containing fires, (**b**) the average depth of chloride migration in mm on tested specimens that were axially split into two pieces, formation of white AgCl product on (**c**) SCC-0M, (**d**) SCC-10M, and (**e**) HySCC-10M.

**Figure 8 materials-15-05588-f008:**
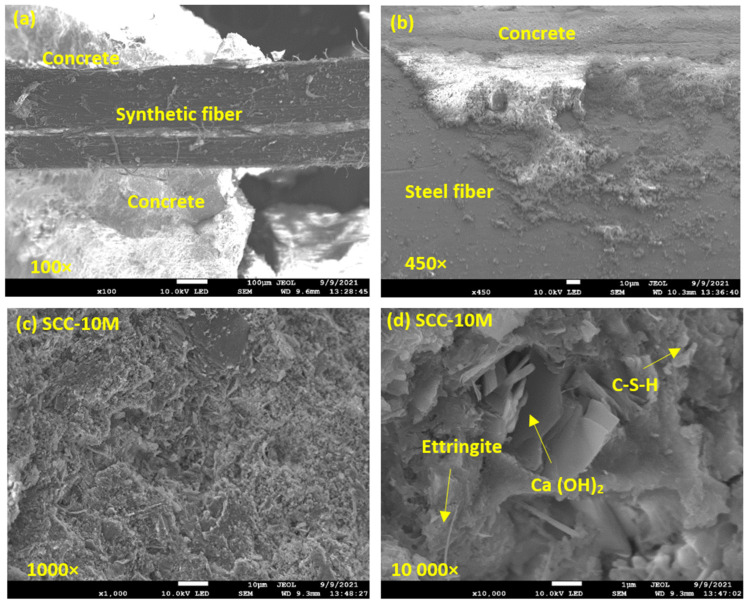
The SEM microstructure (**a**) bonding between fractured SCC samples fixed with synthetic fibers, (**b**) bonding between fractured SCC samples fixed with steel fibers, (**c**) SCC-10M sample at 1000×, and (**d**) SCC-10M sample at 10,000× magnification with corresponding hydration crystallization products.

**Figure 9 materials-15-05588-f009:**
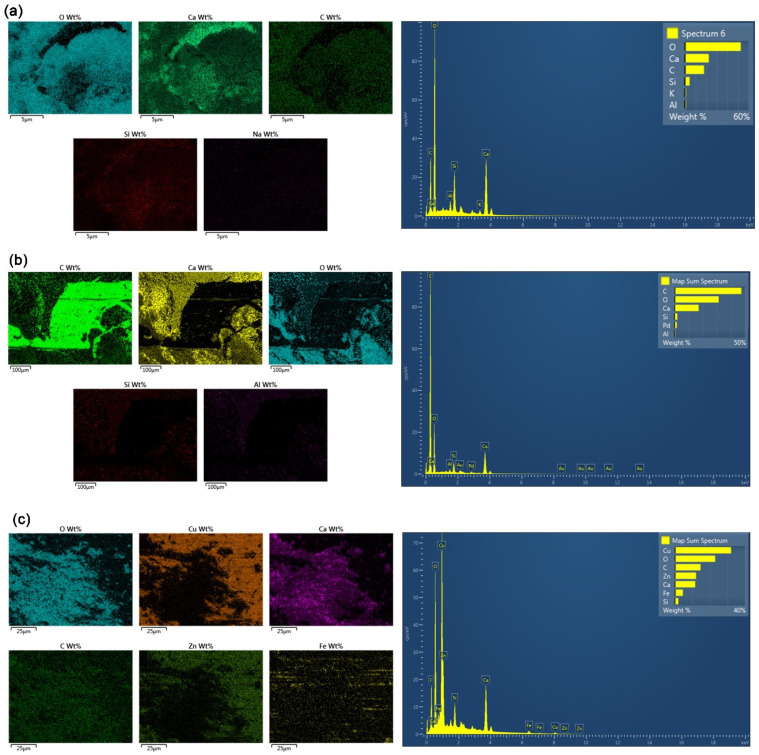
Elemental composition mapping of (**a**) HySSC-10 M sample, (**b**) interface between synthetic fiber and concrete matrix, and (**c**) interface between steel fiber and concrete matrix.

**Table 1 materials-15-05588-t001:** Physical and chemical properties (% by mass) of cement and metakaolin.

Physical Properties			Chemical Properties		
	Cement	Metakaolin		Cement	Metakaolin
Initial setting time (min)	160	120	SiO_2_	16.9	56.0
Density (kg/m^3^)	3000	2600	Al_2_O_3_	4.3	41.0
Specific surface-Blaine (cm^2^/g)	3307	24,000	CaO	62.2	0.3
Particle size distribution (d_50_) (µm)	19.45	4.5	MgO	1.8	0.2
			SO_3_	2.7	-
			Fe_2_O_3_	2.5	1.0
			K_2_O	0.7	0.9

**Table 2 materials-15-05588-t002:** Detailed proportions of the SCC mixes.

SCC Mixes	w/b	M	Cement	FA	CA	Water	SP	VMA	Synt. F	Steel F
	(/)	(kg/m^3^)								
SCC-0M	0.4	0	500	1051	566	200	10	3	/	/
SCC-5M	0.4	25	475	1051	566	200	10	3	/	/
SCC-10M	0.4	50	450	1051	566	200	10	3	/	/
SCC-15M	0.4	75	425	1051	566	200	10	3	/	/
HySCC-0M	0.4	0	500	1051	566	200	10	3	1	50
HySCC-5M	0.4	25	475	1051	566	200	10	3	1	50
HySCC-10M	0.4	50	450	1051	566	200	10	3	1	50
HySCC-15M	0.4	75	425	1051	566	200	10	3	1	50

w/b: water-to-binder ratio, M: metakaolin, FA: fine aggregate, CA: coarse aggregate, FA/CA ratio is 0.65/0.35, SP: superplasticizer, VMA: viscosity-modifying agent, Synt. F: synthetic fibers, Steel F: steel fibers.

## Data Availability

Not applicable.

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
