# Peer review of "Combined Effects of Metakaolin and Hybrid Fibers on Self-Compacting Concrete"

_materials, 2022, doi:10.3390/ma15165588_

Round 1
Reviewer 1 Report
The reviewed manuscript entitled “Combined effects of metakaolin and hybrid fibers on rheological, mechanical, morphological, and durability properties of self-compacting concrete” investigates how both metakaolin hybrid steel and micro-synthetic fibers affect the rheological, mechanical, and chloride ion penetration properties of high-strength self-compacting concrete. The article is made at a good scientific and technical level, and its practical significance is beyond doubt. In order to improve the readability and clarity of the manuscript, some major concerns need to be addressed before the paper is to be accepted for publishing:
1- Discussion is lack of scientific explanation for the obtained results. Authors should attribute the results achieved to a clear scientific reason. Figure 3d is an example, where a random trend is shown and a scientific explanation should be provided to attribute the fluctuation in density for the SCC sample (2300 – 2340 – 2300 – 2340 kg/m3) by changing metakaolin percentages.
2- Figure 5: It is necessary to clarify the indication of the letters A to E on the figure and inside the text as well. Same for figure 7.
3- Figure 10d: The printed magnification (500X) is not matching the original magnification (450X) on the image. Furthermore, the original magnification was removed from image 10c, please replace with the original image.
4- The English language used in the paper is to be revised and improved before the subsequent manuscript submission. Please, read the text carefully before the next submission of the paper.
Reviewer 2 Report
The authors attempt to present the effect of metakaolin and hybrid fibers on self-compacting concreting. The manuscript is well written and will be a good contribution to the scientific community. This manuscript can be accepted subjected to the following clarifications.
1. The title of the manuscript is too long, and contains many properties name. Please relook the tile and make it concise and effective.
2. "The development of self-compacting concrete (SCC) in combination with pozzolans and fibers is essential, as such concrete is expected to have both good mechanical and durability properties." this statement do not support the inclusion of the fiber. Please relook it carefully.
3. Abstract should have one or two lines on the problem statement. it is completely missing. Please rewrite and make it a more comprehensive abstract.
4. once you have defined Self-compacting concrete (SCC) then please use SCC in the remaining part of the manuscript. We used abbreviations to use in short form.
5. The manuscript lacks the clarity to use fiber in the self-compacting concrete. The use of fiber will affect the workability of the SSC, therefore the proper reasoning in the introduction along with the requirement to use fiber should be justified.
6. "Table 1 summarizes the physical and chemical properties of the PC and mineral additive used in the studied mixtures.", what do you mean by PC, it should be defined before.
7. Table 2: please write the units of each material.
8. How you have selected synthetic fiber and steel fiber content. why variation in the content is not considered. The aspect ratio of the fiber plays a vital role in fiber mixing. This issue should be explained clearly. This si the biggest weakness of the manuscript. why only a single aspect ratio has been used?
9. How uniform mixing of the fiber is ensured in the concrete. Howe fibers are added to the concrete. Is any uniformity test been performed on hardened concrete?
10. Why figure 1 and 2: has no meaning. Please remove. Just putting machine pics will not help
11. "3. Results and discussion"; this section has been started as "In this chapter," I think this is copied from the thesis, and please avoid the word CHAPTER. instead, you can use SECTION
12. Figure 4; the maximum values will not help readers to learn the behavior of the materials. It is advised to plot the detailed curves.
13. Figure 10 shows the SEM images of the material; these details are already explained in the literature. either this section should move to the material section, or the concrete mix images should be presented with a detailed explanation.
14. in conclusion, please add the limitation of this study and the practical application of this modified material.
15. Explain SEM and EDX sample preparation methods.
Reviewer 3 Report
After reviewing the paper entitled “Combined effects of metakaolin and hybrid fibers on rheological, mechanical, morphological, and durability properties of self-compacting concrete” I decided to reject it.
The paper has not the quality enaugh to be published in your journal.
The state of the art does not include sufficient evidence to justify the combination of metakaolin and fibers in SCC.
The experiments carried out would have been more appropriate if authors had also made reference concretes only with fibers and with metakaolin. In this way the combined effect of both would have been demonstrated. In the document is not possible to discuss what is the contribution of them in the SCC. In fact, the the microscope analysis results have not been able to prove it.
Finally, references used are quite outdated. None of them are from 2022, only 3 of them are from 2021 and 2 of them are from 2020.
Round 2
Reviewer 1 Report
The revision is satisfactory and the authors have provided amendments to all the suggested queries. The paper has now significantly improved. Therefore, I recommend this work for publication in Materials Journal.
Reviewer 2 Report
The author (s) put minimal effort into addressing the reviewer's comment. The answers have been mentioned just to complete the review process requirement. This study lack a serious research methodology and the basic concept of using the fiber is missing.
This statement shows the limited knowledge related to using any materials in the concrete
"The authors are aware that the properties of fiber reinforced concrete depend on many factors, such as fiber type and aspect ratio (length/diameter), volume percentage of used fibers, fiber orientation and distribution in the concrete matrix and etc. Hence, the following explanation is added to the manuscript in section 2.2. The main aim of this experimental investigation was to study the combined effects of M and hybrid fibers on the various properties of SCC and not to find the optimum content of hybrid fibers which result in the maximum improvement of mechanical properties. For this reason, the amount of synthetic and steel fibers was selected according to the recommendations given in engineering practice. Additionally, to better capture the influence of the combined effects of metakaolin and hybrid fibers on various properties, all other mix components, except the amount of M, were kept constant. Therefore, only one volume fraction of synthetic and steel fibers was considered."
without knowing the optimum size of the fiber, just selecting any random number will not help the research community. Strong reasoning is needed to be explained in the manuscript related to the fiber selection.
How 7 number conclusion can be considered a strong point, it should have been written as a closing remark.
It was asked in the review process to mention the limitation of this study, this point has been suitable ignored.
It was asked about the preparation method of the uniform fiber mixing, however, no satisfactory answer has been found.
This review strongly requests that the author(s) have a look at all the comments of the last report and this report and revise accordingly. At this stage, major revision is recommended.
Reviewer 3 Report
Reject